# Illumination Adaptation in a Multi-Wavelength Opto-Electronic Patch Sensor

**DOI:** 10.3390/s20174734

**Published:** 2020-08-21

**Authors:** Liangwen Yan, Yue Yu, Sijung Hu, David Mulvaney, Panagiotis Blanos, Samah Alharbi, Matthew Hayes

**Affiliations:** 1School of Mechatronic Engineering and Automation, Shanghai University, Shanghai 200444, China; lw_yan@shu.edu.cn (L.Y.); yy_sherlock@shu.edu.cn (Y.Y.); 2School of Mechanical, Electrical and Manufacturing Engineering, Loughborough University, Loughborough LE11 3TU, UK; d.j.mulvaney@lboro.ac.uk (D.M.); P.Blanos@lboro.ac.uk (P.B.); S.M.Alharbi@lboro.ac.uk (S.A.); 3Evonetix Limited., 9a Coldham’s Business Park, Norman Way, Cambridge CB1 3LH, UK; matthew.hayes@evonetix.com

**Keywords:** multi-wavelength opto-electronic patch sensor, heart rate, adaptive algorithm, photoplethysmographic

## Abstract

In capturing high-quality photoplethysmographic signals, it is crucial to ensure that appropriate illumination intensities are used. The purpose of the study was to deliver controlled illumination intensities for a multi-wavelength opto-electronic patch sensor that has four separate arrays each consisting of four light-emitting diodes (LEDs), the wavelength of the light generated by each array being different. The study achieved the following: (1) a linear constant current source LED driver incorporating series negative feedback using an integrated operational amplifier circuit; (2) the fitting of a linear regression equation to provide rapid determination of the LEDs driver voltage; and (3) an algorithm for the automatic adjustment of the output voltage to ensure suitable LED illumination. The data from a single centrally-located photo detector, which is capable of capturing all four channels of back-light in a time-multiplexed manner, were used to monitor heart rate and blood oxygen saturation. This paper provides circuitry for driving the LEDs and describes an adaptive algorithm implemented on a microcontroller unit that monitors the quality of the photo detector signals received in order to control each of the individual currents being supplied to the LED arrays. The study demonstrated that the operation of the new circuitry in its ability to adapt LED illumination to the strength of the signal received and the performance of the adaptive system was compared with that of a non-adaptive approach.

## 1. Introduction

Reflectance photoplethysmography (PPG) is a low-cost non-invasive technique for measuring dynamic changes in the optical properties of living tissue and is becoming increasingly popular in commercial real-time health monitoring products. It has been used to measure oxygen saturation (SpO2), heart rate (HR), blood pressure (BP) and cardiac output (CO) as well as provide an assessment of autonomic function and for detecting peripheral vascular disease [1]. The ZenPPG device, configured with two red LEDs and two infrared LEDs, was developed to monitor blood perfusion in tissue [2]. A second configuration with two red LEDs and four infrared LEDs was able to acquire PPG signals and provide a measurement of SpO2 [3]. Ruiz et al. [4] developed a PPG prototype for the study of heart rate variability (HRV) using reflected green light. Asada et al. [5] designed a ring sensor which was shown to be capable of reliably monitoring patients’ HR, HRV and SpO2. Patterson et al. [6] developed a flexible reflective ear-worn PPG sensor platform for HR monitoring. Wu, W et al. [7] used a broadband light source and color filters to determine pulse rate and arterial blood oxygen saturation. Hong, S et al. [8] selected the sole of the foot as the sampling site rather than the finger or the wrist and optimized the sensor position using LED-PD pairs. Hernando, A et al. [9] showed that a forehead Pulse Amplitude Variability (PAV) signal has a non-respiratory component that was unable to estimate respiratory rate (RR). With the advent of wearable biosensors, few approaches to the monitoring of in vivo physiologic information have been developed, including smartwatches, the Biowatch [10], smart shirts and intelligent finger-rings [11].

To provide a reliable illumination of the tissue for PPG measurements, several authors have recognized that appropriate control of the LED current is needed, and several circuits have been proposed that are able to realize constant-luminance control of LEDs. An LED drive circuit consisting of a boost-type DC-DC converter with constant-output current control and constant-luminance control was proposed by Nishikawa et al. [12]. This circuit is a switching regulator-type circuit controlled by pulse width modulation. To obtain accuracy of the LED current, a constant current source applied to an LED driver was studied by Mu et al. [13]. Hsieh et al. [14] described an offset calibration technique to improve both current accuracy and power efficiency while keeping the chip area relatively small. Chen et al. [15] developed an LED driver with a novel constant current control mechanism that incorporated a charging and discharging module and Wang et al. [16] described a 16-bit constant-current LED driver integrated circuit. Hu Y et al. [17] proposed an LED driver circuit consisting of a voltage pre-regulator and multiple linear current regulators with an adaptively-controlled supply voltage. Zhang J et al. [18] proposed a simple and precise lossless passive current balancing circuit for a multiple LED string application. Wu X et al. [19] proposed a simple rectifying circuit for an isolated two-channel LED driver without active current-sharing control. Saadeh, W et al. [20] proposed a digital controlled low-power LED driver using Pulse Density Modulation (PDM). Atef, M et al. [21] applied Pulse-Width Modulation (PWM) to control the LED and reduce the optical power requirement. Since current imbalance should be avoided among the parallel LED strings, Luo, Q et al. [22] proposed a multi-channel constant current LED driver based on a high-frequency AC bus. To balance the multioutput LED drivers with precise passive current, Zhao, C et al. [23] proposed an improved passive current balancing method based on a capacitive scheme. Sun, T.P et al. [24] proposed a novel light-emitting diode (LED) driver circuit that stabilized the light output without a photodiode. Hu, Y et al. [25] proposed an LED driver circuit consisting of multiple linear current regulators and a voltage pre-regulator with an adaptive output voltage.

The absorption of the light emitted by the LEDs used in PPGs is affected by the factors of skin pigmentation, skin composition, the measurement site and movements of the subject. Consequently, the most appropriate wavelength of illumination and intensity of illumination will generally need to be determined for each subject. A multi-wavelength opto-electronic patch sensor (mOEPS) already has the ability to generate measurements for a set of four alternative wavelengths, which gives the possibility of automatically adapting the system so that it obtains readings only at those wavelengths that provide the best quality signal for each individual subject [26,27].

This study further improved the mOEPS adaptability by adjusting the intensity of the illumination provided by the LEDs. To achieve this, previous published work on providing a suitable constant current was adapted so that a variable current, and hence controllable illumination, can be provided to adapt to the absorption characteristics of an individual’s tissue [27]. This study aims were: (1) to produce a linear constant current source designed to drive the LEDs; (2) to specify linear regression equations fitted for the currents and voltages supplied to the LEDs; and (3) to implement an algorithm for the automatic adjustment of the output voltage (AAOV).

## 2. Implementation of Multi-Wavelength Opto-Electronic Patch Sensor

This section describes the mOEPS system, the method used in the multiplexing of the LED signals, the circuitry used to control the current driving the LEDs and the method used to automatically adjust the intensity of the illumination.

### 2.1. Overview of mOEPS Operation

The mOEPS (Figure 1) measures 18 mm × 18 mm × 0.1 mm and includes 16 LEDs (JMSIENNA Co., Ltd., MiaoLi, Taiwan) as illumination sources [26]. The LEDs are configured in four channels, each channel containing four LEDs and the peak wavelengths of the channels are 525 nm (green), 590 nm (orange), 650 nm (red) and 870 nm (infra-red), and the returned light is captured using a silicon photo diode (PD) with an active area of 1.69 mm^2^ (Hamamatsu photonics K. K., Japan). The printed circuit board (PCB) routing and footprints were designed using the electronics design software platform PADS (Mentor Graphics Co., Wilsonville, OR 97070, USA). A layer of clear epoxy medical adhesive was applied to protect the optical components. Figure 2 shows photographs of mOEPS both with and without the illuminated LEDs.

The required LED intensity is provided from voltage output provided from a PIC-based microcontroller (MCU) (Microchip Technology Inc. Chandler, Arizona, 85224-6199, USA). This voltage is converted by the LED driver to a suitable current to supply to the individual LEDs of the mOEPS, as shown in Figure 3. The signal reflected from the subject is passed through a pre-amplifier, a differential amplifier and a low pass filter to reduce high-frequency noise content such as the power supply frequency. An analog-to-digital convertor (ADC, USB-6009, National Instruments Inc, Austin, TX, USA) is then used to supply a suitable signal for analysis by the MCU. The acquired data are sent to the PC where appropriate processing is carried out using Labview (National Instrument Inc. Austin, TX, USA) to extract the required vital signs. The acquired signals typically contain a direct-current (DC) offset which is removed by a high-pass filter, and the remaining alternating current (AC) components are used in the automated determination of appropriate LED intensities and the required LED driver amplification.

### 2.2. Multiplexing Four Wavelength Illuminations and Demultiplexing Mixture of Received PD Signals

To illuminate the four LED channels in sequence, a method is required to provide time-division multiplexing of the individual LED driver signals as well as ensuring that the signal being received at the PD can be correctly identified with the specific LED being driven.

A time-division multiplexing algorithm was implemented on the MCU to illuminate the LEDs individually in a strict sequence using a frequency of 1 kHz. Figure 4 shows the timing diagram used in the driving of the LEDs. Note that, at every fifth clock pulse, none of the LEDs are driven, thereby allowing capture of information resulting from only ambient lighting conditions and providing immunity from optical interference. Following the sequence of LED illumination, the MCU controls the four-channel sample-hold circuitry to separate the original signals of the LEDs into separate channels. The MCU generates all the pulse signals, co-ordinates the acquisition of the channels captured by the ADC and identifies these as originating from the PD signals captured for specific LEDs. Demultiplexing received signals from the PD extracts each specific wavelength illumination response from the combined signal by isolating one time period of the PD. After cancelling the effect of ambient light, the signals from these individual LED illuminations are acquired, respectively, and passed through a low-pass filter (LPF) (fc = 15 Hz), after which they are converted to four channels of digital signals.

### 2.3. LED Driver

The digitally-controlled LED drivers are an essential part of ensuring the normal operation of the system. Imprecise control of the electrical current would increase switching noise caused by the circuitry. Thus, precise control of the LED drivers and the independent driving of the devices is necessary. Unfortunately, the LEDs used in most commercial probes do not provide a means to ensure a specific optical power, but only ensure a constant brightness by controlling the electrical current supplied to the devices. Optical power control can be achieved by biasing the LED with a resistor, but this method can only provide a constant current if the LED voltage drop and the supply voltage are constant, which limits the LED optical power. A more stable solution is to provide an active circuit that delivers a constant and predetermined current to the LED, thus ensuring the current through the LED is maintained regardless of the LED drop and this is also an important factor for driver stability. Such an active current control methodology is shown in Figure 5, which includes a group of four LEDs whose load current is independent of the voltage drop. This not only ensures constant LED power under switching conditions, but also makes it possible to drive a wider range of devices without modification.

In Figure 5, the LED driver is a linear constant current source, which is composed of a series negative feedback of integrated operational amplifier circuit, including current limiting resistor to limit the maximum output current to protect LED [28]. The circuit is based on the relationship between the input voltage, $v_{1}$ and the current sensing resistors $R_{3}$, $R_{8}$ and $R_{10}$; $R_{10}$ is a current limiting resistor. The relationship between $v_{1}$ and $R_{3}$ determines the current that flows through the first stage. The current gain from the first stage to the second stage is based on the relationship between $R_{8}$ and $R_{10}$.

In the first stage, $v_{1}$ is connected to the non-inverting input of the operational amplifier $U_{5}$. Through the negative feedback, the op amp will control the current through $R_{3}$ so that $V_{3}$ (the ungrounded end of $R_{3}$) is set to the same voltage as $v_{1}$, which is applied to the non-inverting input. The current through the $R_{8}$ resister $I_{8}$ is ideally the same as the current through $I_{3}$. Thus, we can get the voltage at the non-inverting input of the operational amplifier $V_{U1-3}$ through Equations (1) and (2).
(1)$$\;I_{8}=\;I_{3}=\;\frac{v_{1}}{R_{3}},$$
(2)$$V_{U1-3}=VDD-\;I_{8}*\;R_{8},$$

In the second phase, the second amplifier $U_{1}$ is controlled by the voltage drop $V_{8}$ across the $R_{8}$ resistor, which is applied to the non-inverting input. A voltage drop, $V_{10}$, is generated in series by a Darlington tube composed of $R_{10}$, $Q_{3}$ and $Q_{2}$, which is proportional to the load current flowing through $R_{10}$. The $V_{10}$ voltage is applied to the inverting input and the amplifier seta $V_{10}$ and $V_{U1-3}$ to be equal through negative feedback. The current through $R_{10}$ can be calculated through Equation (3).
(3)$$I_{10}=\;\frac{VDD-V_{10}}{R_{10}}=\;v_{1}*\;\frac{R_{8}}{R_{3}*\;R_{10}},$$

When the $V_{U1-3}$ reaches the minimum voltage that $U_{5}$ can output, the voltage drop on the resistor $R_{10}$ reaches the maximum and the circuit output current also reaches the maximum. Thus, the $R_{10}$ resistance determines the maximum current the circuit can output. The triodes $Q_{4}$ and $Q_{6}$ can also protect the circuit. Before the maximum current limit is reached, the total current through the LED can be calculated through Equation (4).
(4)$$I_{LED}=\;I_{10}=\;\frac{VDD-V_{10}}{R_{10}}=\;v_{1}*\;\frac{R_{8}}{R_{3}*\;R_{10}},$$

The bandwidth limiting capacitors $\;C_{5}$ and $C_{3}$ are included to maintain stability of the feedback loop and prevent both oscillations and overshoot in the step response. In reality, for mOEPS, four groups of the same functional driver circuitry are necessary to drive green, yellow, red and infrared LEDs separately.

### 2.4. Automated Adjustment of the LED Intensity

Individual tissue characteristics such as skin pigment, thickness and composition affect the absorption of the illumination provided by the LEDs. If the intensity is too low, then the wanted signal could be too small and become severely affected by noise; conversely, if the intensity is too high, a saturated signal could be received at the PD [29]. Figure 6 shows the algorithm used for the automatic adjustment of the output voltage (AAOV) implemented in the current work. Three metrics in the algorithm works were defined with the desired value and both lower and upper threshold values. These separate sets of values can be determined by each of the operating LED channels. To provide the required current to the LEDs, a curve was found to express the relationship between the output voltage from the MCU and the current supplied to each of the LED channels. The operation was carried out to provide a suitable calibration of current supplied to the LEDs as follows. To obtain AC signal and to avoid LED current changes from any interfering with the analog outputs, the amplitude of the raw signal obtained by the PD needs to keep a constant value within the threshold range. Firstly, to check the designated threshold state, the MCU compares the amplitude of the raw signal received by each channel to check whether it falls below the lower threshold or above the upper threshold. When the level of signal is out of the threshold range, MCU switches to check the desired value state and increases or decreases the voltage that is converted to drive current. Secondly, to check the desired value state, the amplitude of the raw signal is compared with the desired value. This leads an appropriate current value to drive LED to a value, thus resulting in the received signal being close to the desired value of Voltage. After this, the MCU automatically switches back to check the threshold state again. Finally, through these steps, these different LED channels/groups are provided with different levels of driving voltages to obtain the optimal total illumination of each LED groups. Hence, AC signals of each channel can be accurately separated. The total adaptation period takes a period of 2–3 s. After this procedure, AAOV could then remain dormant again to allow further amplitude measurements to be taken until the signal falls outside the thresholds again.

## 3. Results

The experiments were carried out to attain the relationship between the input voltage $v_{1}$ and the output electrical current I through LED. A green LED, a red LED, an orange LED and an IR LED were respectively connected in series with a 10-Ω resistor. The results are shown in Table 1.

The multi-channel LED driver was evaluated experimentally. The relationship between the voltage supplied by the MCU and the current received at the LEDs was established using a least-squares method available in MATLAB (MathWorks Inc., Natick, MA, USA). The current supplied to each of the LEDs channels was verified to be stable throughout the switching processes. To achieve the quick responds of system, the curve of LEDs will be simplified to linear equations. For validation purposes, HR signals were collected for input to the AAOV and these were determined following suitable bandpass filtering and processing using frequency analysis.

For a suitable operation range of the AAOV, the voltages supplied by the MCU to drive each of the LEDs needed to be determined. A set of empirical measurements were taken to provide output currents in the valid range. The measured voltages required to drive each of the LEDs are shown in Figure 7. The experimental results of the relationships between the output currents i obtained at the LED for a range of MCU voltages v, for each of the green, orange, red and infrared LED illuminations are shown. A suitable model fitted with all four curves was found to be an exponential relationship between i and v. The MCU voltage needed to produce a given LED current depends on the LED color. The linear regression equations were fitted over the range of operation, as shown in Figure 7.

Figure 8 shows the signals obtained from one subject while at rest both when AAOV was and was not applied. In Figure 8a, the peak-to-peak waveform amplitude of the signal received from the green LED when not using AAOV is 1.7 V, but, as shown in Figure 8b, when AAOV was implemented, the corresponding waveform amplitude became 2.6 V. This was due to the AAOV system producing a larger current to drive the LEDs. The HR measured when AAOV was not used was 84 bpm, whereas the HR measured when AAOV was applied was 82 bpm. An ECG measurement was taken to provide a golden standard for comparison purposes, which was found to give a value of 81 bpm. Tests carried out on other subjects confirmed the improvement in accuracy that could be obtained by using AAOV.

## 4. Discussion

Two crucial aspects in this study were addressed to make the mOEPS operate regularly: (1) an appropriate intensity of LEDs illumination is requested to ensure the current passed through the LEDs to be well controlled; and (2) the mixture of received backlight signals can be separated from a single PD and demultiplexed from the four wavelength illuminations of LEDs., Two extreme circumstances have to be taken account during the operation of mOEPS. Firstly, the AC components from the signal of PD is not easily extracted from a raw PPG signals due to saturated response of the PD corresponding with an over illumination of LED. Secondly, when the intensity of LEDs illumination is not strong enough to pass through the objective tissue, the signals captured from the PD are too weak to extract any valid AC components. Thus, the multiplexed LEDs driver needs to provide an appropriate current (mA) to an individual wavelength illumination source (LEDs) based on the feedback of the de-multiplexed signal. Hence, the correction between the LEDs’ current and voltage was studied to improve the efficiency of adjusting suitable current through the LEDs.

Table 1 shows that, as the voltage of $v_{1}$ gradually increases, so does the current through LED. Meanwhile, the output electrical current of the control circuit only depends on the input voltage $v_{1}$ regardless of the wavelength of LEDs. Thus, this electrical circuit in Figure 5 realizes constant current source to drive LEDs.

In Figure 7, the linear corrections between voltage and current of LED drivers are: green LEDs, R^2^ = 0.9904; orange LEDs, R^2^ = 0.9993; red LEDs, R^2^ = 0.9745; and infrared LEDs, R^2^ = 0.9885. Hence, the LEDs illumination could be adjusted quickly and individually based on the feedback value of the PD signals.

Figure 8 expresses a comparative test of the pulsatile signals with and without the AAOV applied. The auto-attenuation or auto-increase of LED illumination corresponded with the relevant amplitude variations of pulsatile signals, and the multi-wavelength (four-channel) LED illuminations were controlled to obtain multiplexed pulsatile signals.

In the mOEPS, four wavelength illuminations were selected to measure HRs. It has been observed that the HRs obtained from green wavelength illumination is closer to the gold standard (ECG). Thus, the green wavelength illumination is better to obtain the HR.

## 5. Conclusions

This paper presents new work that includes the design of a PPG system that is able to adapt the illumination of the acquisition site according to the characteristics of the received signals. Firstly, the LED driver circuit was implemented to ensure a linear steady current source which is composed of the series negative feedback of an integrated operational amplifier. This provides an appropriate intensity of LED illumination to suit individual circumstances. Secondly, the linear regression equations are fitted according to the LED driver current, thus producing a range of operation that greatly improves the computing efficiency of the LED driver voltages. Thirdly, an algorithm is used for the automatic adjustment of the output voltage to ensure suitable LED illumination. The adaptation implemented for the mOEPS system not only allows an initial configuration to be made before readings are acquired, but the approach also allows adaptation to be carried out continually while measurements are being taken. This allows the MCU, which is monitoring the quality of the PPG signal being received, to make suitable changes to its output voltage, so that either signals of greater amplitude can be obtained thereby reducing the presence of noise or the driver current is reduced thereby avoiding saturation of the PD.

Further work is being carried out to validate the operation of the AAOV system on a larger trial. At present, the illumination from each LED is considered individually, whereas combinations of different colored lighting may improve the quality of the signals being received. Several challenges remain that are also being considered in follow-on studies, including motion artifact reduction and shielding from electrical noise.

## Figures and Tables

**Figure 1 sensors-20-04734-f001:**
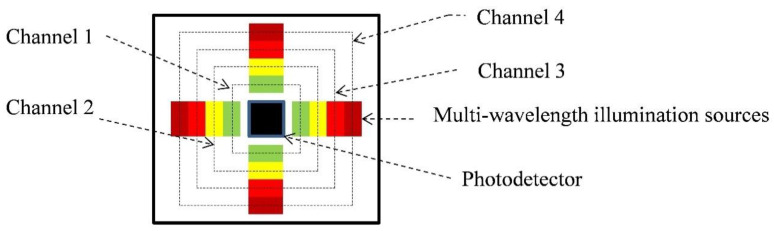
Multi-wavelength opto-electronic patch sensor (mOEPS). Each channel has four LEDs: Channel 1, green LEDs; Channel 2, orange LED; Channel 3 red LEDs; and Channel 4, IR LEDs.

**Figure 2 sensors-20-04734-f002:**
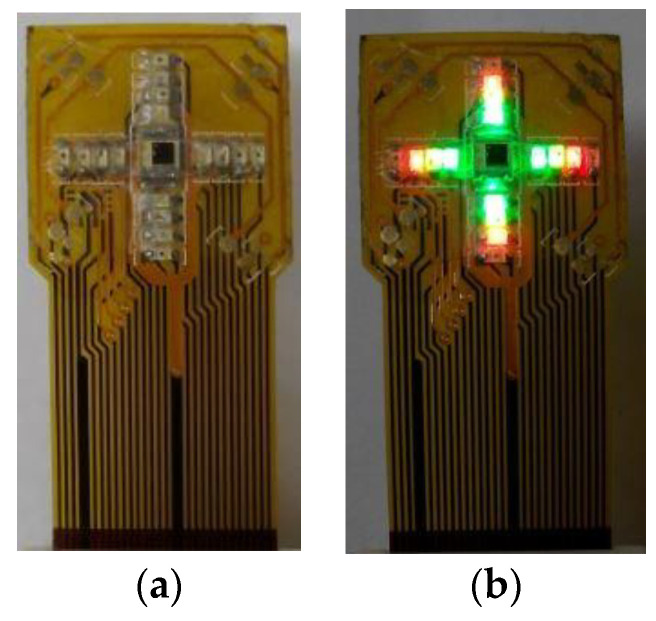
Photographs of the mOEPS: (**a**) in non-illuminated state; and (**b**) in illuminated state.

**Figure 3 sensors-20-04734-f003:**
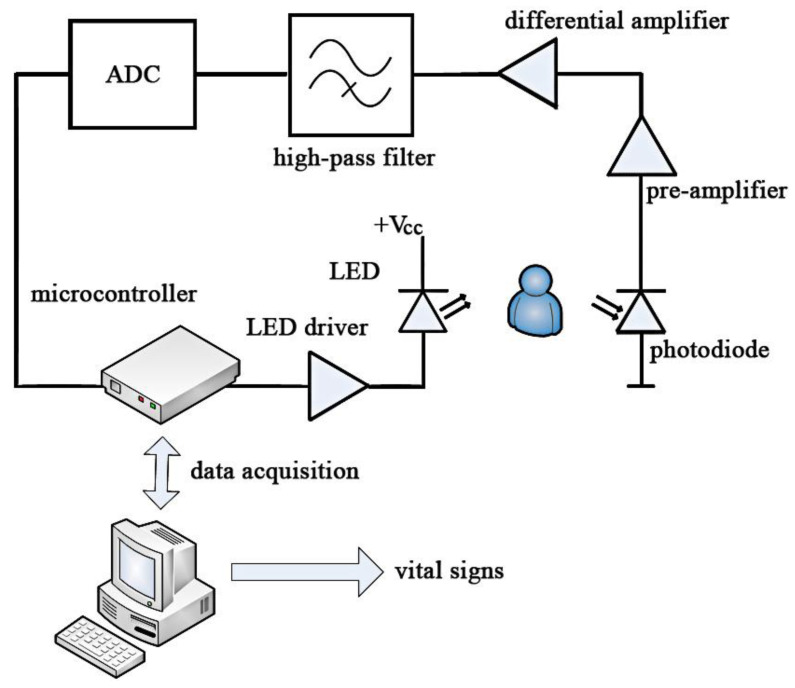
Block diagram of the mOEPS circuit.

**Figure 4 sensors-20-04734-f004:**
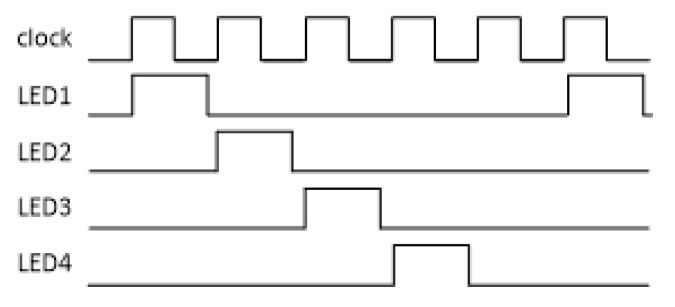
Timing diagram showing the order in which the LEDs are driven.

**Figure 5 sensors-20-04734-f005:**
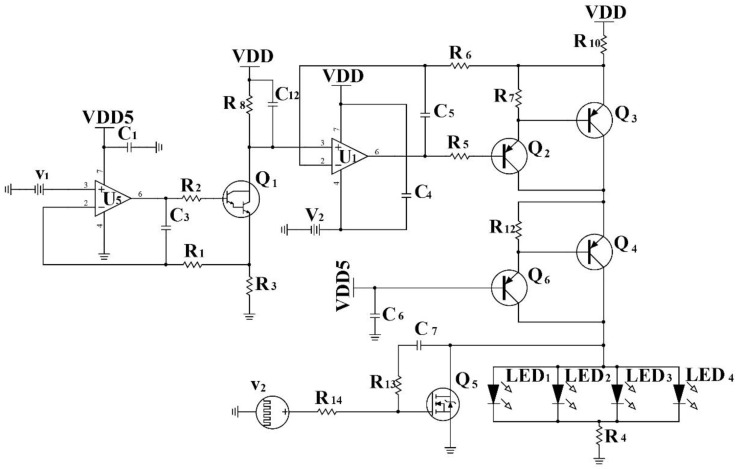
Schematic diagram of LED current control circuit. $U_{1}$ is an amplifier with a non-inverting input supplied from the DAC and the base of $Q_{5}$ is controlled by the MCU to switch the LEDs on or off.

**Figure 6 sensors-20-04734-f006:**
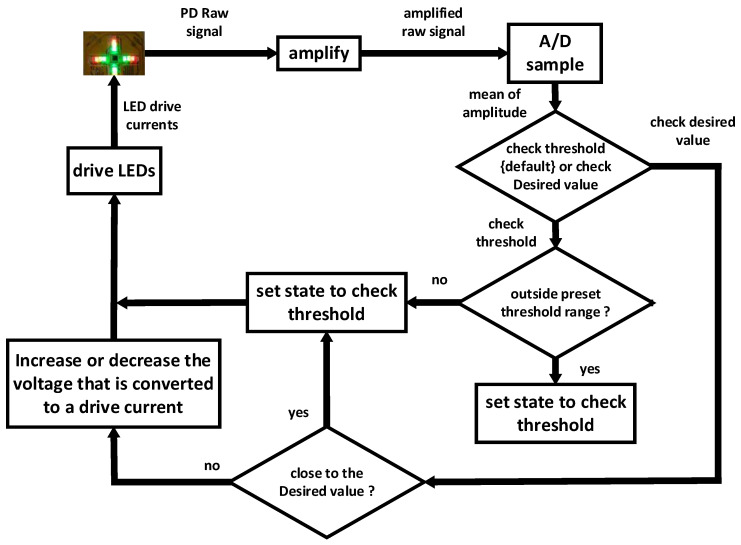
For acquiring high-quality PD signals, the two states of AAOV algorithm are shown in this figure: (1) check the threshold state and determine whether the current intensity of LED needs to be adjusted; and (2) check the desired value state and adjust the LED current to the desire value to obtain an appropriate original signal from PD.

**Figure 7 sensors-20-04734-f007:**
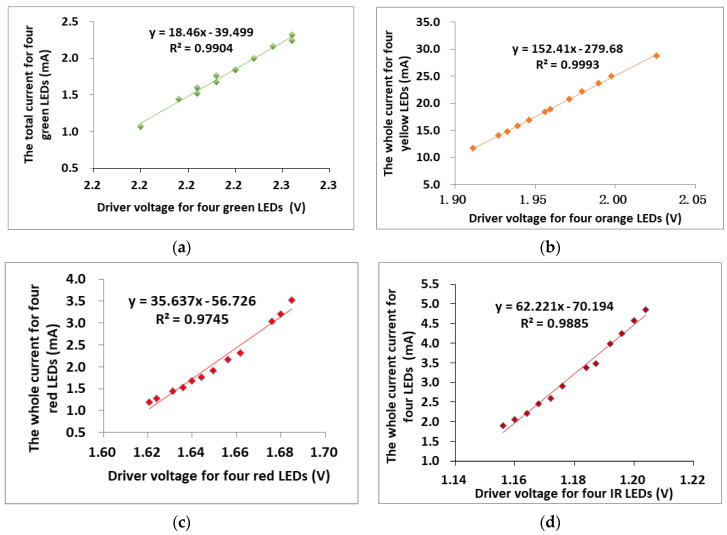
Linear approximations of the relationship between the current supplied to the LEDs and voltage supplied by the MCU. The relationship between the current supplied to the LEDs and voltage supplied by the MCU for the green LEDs (525 nm), orange LEDs (590 nm), red LEDs (650 nm) and infrared LEDs (870 nm): (**a**) green LEDs; (**b**) orange LEDs; (**c**) red LEDs; and (**d**) IR LEDs.

**Figure 8 sensors-20-04734-f008:**
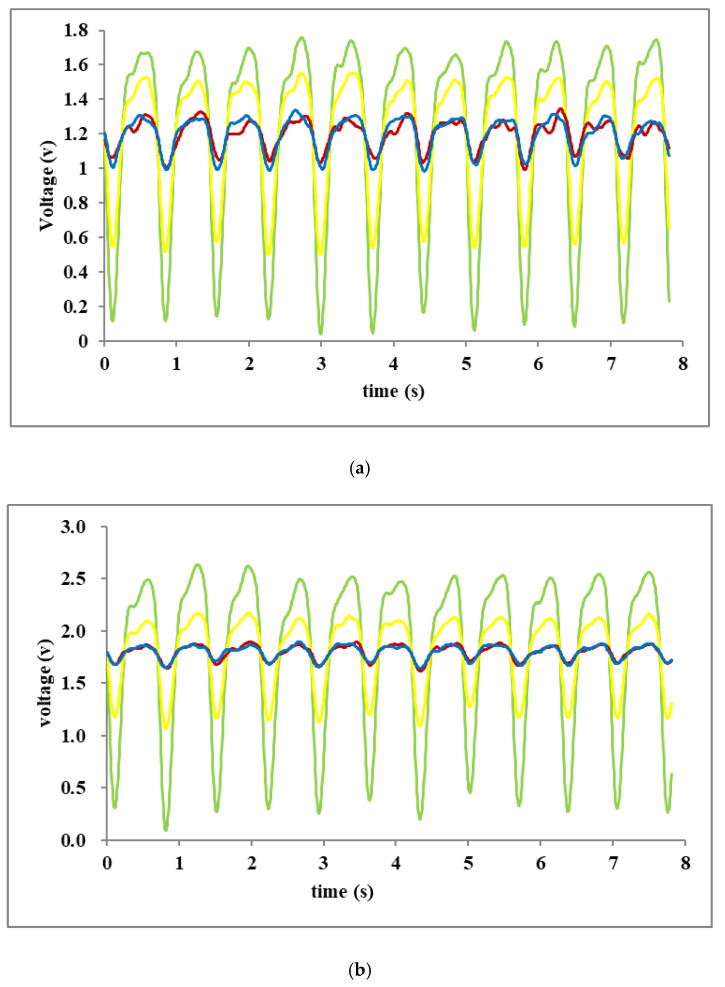
The PD signals received from a subject at rest. Note that green signal is the signal received from the green LED illumination; orange from the orange LED illumination; red from the red LED illumination; and brown from the infrared LED illumination: (**a**) the AAOV not applied; and (**b**) the AAOV applied.

**Table 1 sensors-20-04734-t001:** The relationship between the input voltage $v_{1}$ (V) and the current I (mA) through the designated LED, where VDD is the drain voltage.

$\nu_{1}$	VDD	$U_{1-3}$	$I_{10}$	$I_{GREEN}$	$I_{RED}$	$I_{ORANGE}$	$I_{IR}$
0.02	12.05	12.03	0.67	0.60	0.60	0.60	0.60
0.04	12.05	12.01	1.33	1.30	1.30	1.30	1.30
0.06	12.05	11.99	2.00	2.00	2.00	2.00	2.00
0.08	12.05	11.97	2.67	2.60	2.60	2.60	2.60
0.10	12.05	11.95	3.33	3.30	3.30	3.30	3.30
0.12	12.05	11.93	4.00	4.00	4.00	4.00	4.00
0.14	12.05	11.91	4.67	4.60	4.60	4.60	4.60
0.16	12.05	11.89	5.33	5.30	5.30	5.30	5.30
0.18	12.05	11.87	6.00	6.00	6.00	6.00	6.00
0.20	12.05	11.85	6.67	6.60	6.60	6.60	6.60

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
