# Peer review of "Illumination Adaptation in a Multi-Wavelength Opto-Electronic Patch Sensor"

_sensors, 2020, doi:10.3390/s20174734_

Round 1
Reviewer 1 Report
This work is very interesting and presents a good scientific quality and could be relevant. It is well presented from the point of view of the Methodology and Results. Its publication is recommended.
Just some aspects of improvement that the authors could consider:
The abstract must be rewritten and adjusted exactly to the objective and real results of the work
A more rigorous and updated bibliographic review is recommended in the Introduction part that allows and a better definition of the state of the art in the subject matter under study. The general and specific objectives of the study to be carried out must be indicated more clearly.
It is recommended to introduce a Discussion section where the advantages and disadvantages of an innovative model presented compared to other models are analyzed with critical judgment.
A revision of the conclusions section is recommended. It is very important to better relate the Conclusions with the Study Objectives presented in the introductory part.
The figures must be numbered appropriately, mind (some are repeated as figure 7) as well as improve the quality. They are difficult to observe just as the Flowchart should be better explained and more clearly represented in figure 6
Bibliography is scarce and should be reviewed and updated to improve the quality and interest of readers and researchers
Once these considerations have been made as a recommendation, the work could be published with the approval of the editors.
Author Response
Dear Reviewer 1
Appreciate your valuable comments.
Please see the attachment to answer your questions.
All the best
Sijung Hu on behalf of the list of authors.

Reviewer 2 Report
Summary:
The authors proposed an interesting approach using a multi-wavelength opto-electronic patch sensor (mOEPS) to detect the photoplethysmographic signals (PPG). The idea is using this patch sensor to adapt in an automatic way the illumination intensities of LEDs to improve the quality of the PPG signal.
Broad comments:
The method seems very promising. Overall, the introduction and the presentation of experiments and their results are quite well done. However, this reviewer is suggesting that prior to publication, the authors consider the following minor editorial revisions.
Specific comments:
- In section 1.(Introduction) the authors should extend the description of mOEPS sensor. The reader does not anything about this patch sensor, so it is necessary a brief description of it, if there are some papers about it you can quote it. In your case is a self-citation but personally I think that is necessary for this paper that the authors quote the prior works. Moreover, in row 54 the authors write "mOEPS" without specifying whats mean. In row 59 the authors write "previous published work" it is necessary the reference.
-
Some abbreviations are also missing. Furthermore there are many abbreviations, for this reason, I suggest to the authors to report a short table with a short description of each term (mOEPS, PCB, PD, VDD, MCU, LPF...)
- The following typographical error was detected in line 78: LEDs,. remove: ",".
- The figure 5 is not clear. I suggest to the authors to change the font size.
- In section 3. In line 187 and 188 the authors use "v_1" and "I". It would be better if the authors use the italics font like the previous paragraph (The same things for line 203 and 204). Moreover about this section, Which type of PPG sampling device has been used? Add details. Which kind of pipeline has been used to stabilize/filter the raw PPG signals with respect to artefacts (body movements, etc..) or electronic noise? Add more details about the above questions if the authors consider it would be useful. Furthermore, in line 208 the authors write: "Their linear regression equations are as follows:" but the sentence is incomplete.
- In line 223 is reported "Figure 7" instead "Figure 8". About this figure, I suggest to the authors to change the brown colours of the graph (the brown and red lines are mixed).
In general, the manuscript is reasonably well done and close to being in the appropriate form for future publication, so it is recommended that the
authors consider some of these minor revisions and resubmit the revised manuscript for future publication.
Author Response
Dear Reviewer 2
Appreciate your valuable comments.
Please see the attachment to answer your questions.
All the best
Sijung Hu on behalf of the list of authors.

Reviewer 3 Report
This paper provides circuitry for driving the LEDs and describes an adaptive algorithm implemented on a microcontroller unit that monitors the quality of the PD signals received in order to control each of the individual currents being supplied to the LED arrays. On the whole, the theoretical innovation is weak, but it is acceptable。References should be updated.Author Response
Dear Reviewer 3
Appreciate your valuable comments.
Please see the attachment to answer your questions.
All the best
Sijung Hu on behalf of the list of authors.

Reviewer 4 Report
The paper discuss about a design of a multi-wavelength multi-channel patch sensor for obtaining PPG signals. The approach of the multi-channel LED system with individual current control as per needs is interesting for future research work.
please see the following comments and address them.
- Line 88: LabVIEW is from National Instruments, NOT from TI.
- Figure 1. : Some dotted lines for different channel are not visible.
- Section 2.2: Explain in detail how you derive different channel signals from the detector signal.
- Figure 5: Some labels are not legible.
Author Response
Dear Reviewer 4
Appreciate your valuable comments.
Please see the attachment to answer your questions.
All the best
Sijung Hu on behalf of the list of authors.
